# REFINING CORPORA FROM A MODEL CALIBRATION PERSPECTIVE FOR CHINESE SPELLING CORRECTION

## ABSTRACT

Chinese Spelling Correction (CSC) commonly lacks large-scale high-quality corpora, due to the labor-intensive labeling of spelling errors in real-life human writing or typing scenarios. Two data augmentation methods are widely adopted: (1) *Random Replacement* with the guidance of confusion sets and (2) *OCR/ASR-based Generation* that simulates character misusing. However, both methods inevitably introduce noisy data (e.g., false spelling errors), potentially leading to over-correction. By carefully analyzing the two types of corpora, we find that though the latter achieves more robust generalization performance, the former yields better-calibrated CSC models. We then provide a theoretical analysis of this empirical observation, based on which a corpus refining strategy is proposed. Specifically, OCR/ASR-based data samples are fed into a well-calibrated CSC model trained on random replacement-based corpora and then filtered based on prediction confidence. By learning a simple BERT-based model on the refined OCR/ASR-based corpus, we set up impressive state-of-the-art performance on three widely-used benchmarks, while significantly alleviating over-correction (e.g., lowering false positive predictions).

## 1 INTRODUCTION

Chinese Spelling Correction (CSC) aims to detect and correct misspellings in the text while maintaining the sentence length Yu & Li (2014). It can not only directly facilitate human writing and typing but also serve as a critical pre-processing step for many downstream Chinese NLP tasks such as search engine Martins & Silva (2004) and optical character recognition Afli et al. (2016). One common challenge of applying CSC is the lack of large-scale high-quality corpora in practice since labeling spelling errors in real-life writing or typing scenarios is labor-extensive Wang et al. (2018). Therefore, two data augmentation methods are widely adopted for this task. The first one is *random replacement* with the guidance of confusion sets Liu et al. (2013) containing typical human misused cases based on statistics. The second one is leveraging cross-modal models Wang et al. (2018), such as optical character recognition (OCR) and automatic speech recognition (ASR), to simulate spelling errors in the shape-close or tone-close patterns.

Compared to random replacement, OCR/ASR-based generation better mimics human misspelling scenarios, becoming the mainstream strategy used by many recent CSC efforts Cheng et al. (2020); Wang et al. (2021). Unfortunately, both data augmentation methods inevitably introduce noises. For example, we randomly sample 300 sentences in the OCR/ASR-based corpus Wang et al. (2018) and check the annotated misused characters manually, finding that 11.3% of them are false spelling errors. Training on these noisy samples can produce unintended over-correction (e.g., a high false positive rate). Previous works mainly alleviate the problem through sophisticated model designs, e.g., integrating phonological and morphological information using multi-modal approaches Xu et al. (2021); Huang et al. (2021). Unlike these efforts, in this paper, we propose to improve CSC by directly purifying noisy samples in CSC corpora.

Considering model confidence is commonly exploited to denoise data Northcutt et al. (2021), we first analyze the two types of CSC corpora by checking the calibration characteristics and performance of models trained on them (see Section 2 for experiment details). The experimental results on the SIGHAN 13 Wu et al. (2013) benchmark are shown in Figure 1 [1]. Comparing subplots (a) and (b),

---

[1] Appendix B shows the results on SIGHAN 14/15

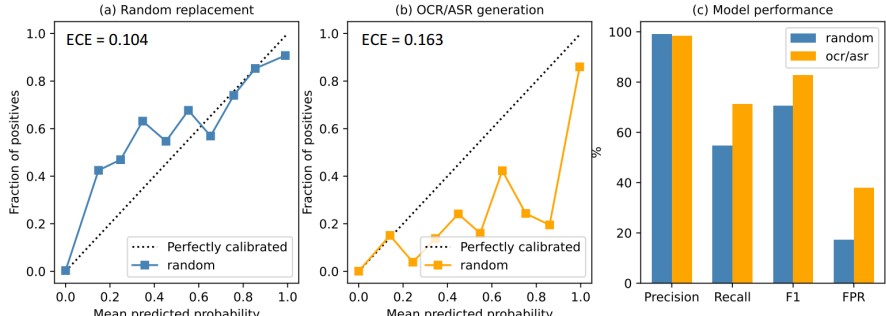

Figure 1: Calibration curves and performance of BERT-based CSC models trained on random replacement and OCR/ASR-based data. ECE refers to the Expected Calibration Error metric Guo et al. (2017), and FPR represents the sentence-level false positive rate, which measures over-corrections. Combining subplots (a), (b), and (c), OCR/ASR-based data demonstrate superior performance on standard metrics such as precision (P), recall (R), and F1 score. However, random replacement data exhibit better calibration and lower FPR.

we find that although the CSC model trained on OCR/ASR-based data performs better (e.g., with a better F1 score), it is worse calibrated than its counterpart of random replacement. Its calibration curve continuously lies below the dotted line (representing perfectly calibrated), indicating that the model tends to make over-confident predictions. This observation is consistent with its higher false positive rate (despite overall better performance) in subplot (c). To explain the empirical observation, we then perform a theoretical analysis of model confidence based on bayesian inference (Section 3). We reveal why the calibration curve differs between the two categories of training data and identify which data samples negatively affect model confidence.

Guided by the empirical observations and theoretical findings, we propose to refine the OCR/ASR-based corpus with a CSC model trained on random replacement data. Thanks to this CSC model's more trustful confidence, we can use it to filter noisy OCR/ASR-based samples according to their prediction scores. We achieve competitive performance on three open CSC benchmarks by training a simple BERT-based model on the refined corpus. Notably, the model also produces a much lower false positive rate and demonstrates better calibration, which is essential in real-world CSC applications.

In summary, our contributions are as follows:

- We empirically reveal that OSC/ASR-based CSC datasets deliver more robust generalization performance, while random replacement datasets lead to better-calibrated models.
- We theoretically analyze models' calibration characteristics from a bayesian inference view, explaining how and which data samples bring the unintended over-confidence of predictions.
- We design a corpus refining strategy that integrates the generalization performance from OSC/ASR-based data and the trustful model confidence from random replacement data.

## 2 A PILOT STUDY OF DATA CHARACTERISTICS

Figure 1 illustrates the properties of OCR/ASR-based and random replacement data through the calibration curves and performance of their respective models. The Expected Calibration Error (ECE) metric is explained in detail in Appendix A. In this section, we provide a comprehensive description of the experimental methodology and procedures.

### 2.1 THE BASE CSC MODEL

Given data pair $(X, Y)$, where $X$ is the original sentence and $Y$ is the generated sample containing spelling errors, Chinese spelling correction aims to restore $Y$ to $X$. Since $X$ and $Y$ share the same sentence length, this task is usually implemented by a non-autoregressive model. In this work, $Y$ is input into a BERT model, and the output hidden state of each character is fed into a classifier to get the predicted correct character. The training target can be written as the following cross-entropy loss:

$$L_{CE} = -\sum_{i=1}^{L} log[P_\theta(x_i|Y)] \tag{1}$$

where $L$ is the shared length and $\theta$ represents model parameters.

## 2.2 ANALYSIS OF TWO DATASETS

**Dataset Preparation**. We use the OCR/ASR-based dataset containing 271k sentences provided by Wang et al. (2018). We can build a confusion set based on its annotated spell errors. To obtain a random-replacement dataset of similar volume, we collect the same number of sentences and then uniformly substitute correct characters with a probability of 10% with characters in the constructed confusion set. In this way, we can compare two types of datasets fairly.

**Metrics Settings**. Regarding model performance, in addition to standard metrics (e.g., precision (P), recall (R), and F1), we also examine sentence-level false positive rate (FPR) Li et al. (2022c). A sentence is regarded as a false positive if any initially correct character is wrongly modified to another one. Regarding model confidence, since most of the characters in the dataset are correct, numerous easy positive samples will blur the noteworthy trends in calibration curves. Therefore, we eliminate those characters—in whose prediction distribution the possibility of being corrected to other characters is below 0.1—to draw the calibration curve and calculate ECE.

**Main findings**. The main results of SIGHAN 13 have been shown in Figure 1, and more experimental results of SIGHAN 14 and 15 are placed in Appendix B due to space limitation. In all three datasets, we can observe in the calibration line chart that the CSC model trained on OCR/ASR-based data is flawed regarding the alignment between prediction confidence and accuracy, despite the better overall performance. ECE scores achieved by random replacement and OCR/ASR-based generation are 0.104 and 0.163, respectively, suggesting that the former is closer to the ideal calibration and also explaining why it achieves a lower FPR (e.g., with fewer over-corrections).

## 3 THEORETICAL ANALYSIS OF MODEL CONFIDENCE

### 3.1 PROBLEM STATEMENT

In this section, we present a theoretical analysis of the above empirical findings. To begin, we define a set $\mathcal{X}$ that each element, denoted as $X = (x_1, x_2, ..., x_L)$, represents a sentence in the real-world corpus comprised of individual characters. The prior probability of the sentence can be determined using the probability function $P_{\mathcal{X}}$. By some methods of data augmentation, a mapping function $\mathcal{F} : \mathcal{X} \rightarrow \mathcal{Y}$ is applied to imitate human's writing error set $\mathcal{Y}$, which consists of sentences containing a small number of incorrect characters. The probability of sentences in $\mathcal{Y}$ is obtained from $P_{\mathcal{Y}}$.

For any sentence $X \in \mathcal{X}$, we assume the mapping function $\mathcal{F}$ replaces only one character at a time. $Y = \mathcal{F}(X), y_i = \mathcal{F}(X)_i \neq x_i$. We denote the context of $x_i$ as $X_{\setminus i} = (x_1, ..., x_{i-1}, x_{i+1}, ..., x_L)$. Based on these assumptions, we can draw the following simple inferences:

- $X_{\setminus i} = Y_{\setminus i}$: This equality implies that the context surrounding the replaced character remains unchanged when transforming $X$ to $Y$.
- $P_{\mathcal{X}}(X_{\setminus i}) = P_{\mathcal{Y}}(Y_{\setminus i})$. Since the data augmentation methods do not alter the size of the dataset, we can assert that $|\mathcal{X}| = |\mathcal{Y}|$. There is a one-to-one correspondence between the contexts in $\mathcal{X}$ and $\mathcal{Y}$. Consequently, we can establish an equation relating the probabilities of $X_{\setminus i}$ and $Y_{\setminus i}$.

### 3.2 BAYESIAN INFERENCE OF MODEL CONFIDENCE

Combining the inferences, we can derive the theoretical correction model confidence $P(X|Y)$ from a Bayesian inference perspective, as the probability $P(Y|X)$ in the augmentation process is known.

$$P(X|Y) = \frac{P(y_i|X) \cdot P_{\mathcal{X}}(x_i|X_{\setminus i})}{\sum_{v \in \mathcal{V}} P(y_i|X_{\setminus i}, v)P_{\mathcal{X}}(v|X_{\setminus i})} \tag{2}$$

In the formulation, the vocabulary $\mathcal{V}$ encompasses all possible characters. The detailed calculation procedure is presented in Appendix D.To further decompose Eq. 2, we define a subset $\hat{\mathcal{V}} \subset \mathcal{V}$, which consists of the characters $v$ that make both $P(y_i|X_{\setminus i}, v)$ and $P_\mathcal{X}(v|X_{\setminus i})$ non-zero.

$\hat{\mathcal{V}}$ satisfying the condition is usually categorized into the following three orthogonal cases. The next section will provide more intuitive explanations of the three cases.

**Case 1:** $|\hat{\mathcal{V}}| = 1$, in other word, $\hat{\mathcal{V}} = \{x_i\}$.

$$P^T(X|Y) = \frac{P(y_i|X) \cdot P_\mathcal{X}(x_i|X_{\setminus i})}{P(y_i|X_{\setminus i}, x_i)P_\mathcal{X}(x_i|X_{\setminus i})} = 1 \tag{3}$$

**Case 2:** $y_i \in \hat{\mathcal{V}}$, for simplicity, let $\hat{\mathcal{V}} = \{x_i, y_i\}$.

$$P^N(X|Y) = \frac{1}{1 + \frac{P_\mathcal{X}(y_i|X_{\setminus i})}{P_\mathcal{X}(x_i|X_{\setminus i})} \cdot \frac{P(y_i|X_{\setminus i}, y_i)}{P(y_i|X_{\setminus i}, x_i)}} \tag{4}$$

**Case 3:** $y_i \notin \hat{\mathcal{V}}$ and $|\hat{\mathcal{V}}| > 1$. To simplify the notation, let $\hat{\mathcal{V}} = \{x_i, a\}, a \neq y_i$.

$$P^M(X|Y) = \frac{1}{1 + \frac{P_\mathcal{X}(a|X_{\setminus i})}{P_\mathcal{X}(x_i|X_{\setminus i})} \cdot \frac{P(y_i|X_{\setminus i}, a)}{P(y_i|X_{\setminus i}, x_i)}} \tag{5}$$

### 3.3 DATA SAMPLE CATEGORIZATION

The three cases discussed in the previous subsection are naturally related to the three sample types in the CSC dataset. Symbolic examples are presented in Table 1. We analyze the impact of different data augmentation methods on these sample types.

| Case | original | replaced | truth set of A?C |
|------|----------|----------|-------------------|
| 1 | ABC | ADC | {B} |
| 2 | ABC | ADC | {B,D} |
| 3 | ABC | ADC | {B,E} |

Table 1: Symbolic illustration of different cases. The characters identified by underscores in the second and third columns correspond to $x_i$ and $y_i$ respectively.

**True Sample** corresponds to Case 1, where the context $X_{\setminus i}$ can determine the unique character $x_i$, or there are multiple suitable characters, but $y_i$ only appears in the confusion set of $x_i$.

**Noisy Sample** corresponds to Case 2. In this case, a correct sentence can unexpectedly be transformed into another correct one during data augmentation, generating false spelling errors.

When considering the four terms in the denominator of Equation 4, regardless of the data augmentation method, $P_\mathcal{X}(y_i|X_{\setminus i})$ and $P_\mathcal{X}(x_i|X_{\setminus i})$ remain the same. Additionally, $P(y_i|X_{\setminus i}, y_i)$ will be close to 1, as misspellings generally constitute only a small percentage of all characters. Therefore, $P(y_i|X_{\setminus i}, x_i)$ is the primary factor influencing $P^N(X|Y)$.

Specifically, random replacement data provide a uniform distribution for $P(y_i|X_{\setminus i}, x_i)$, which can stabilize $P^N(X|Y)$. On the other hand, OCR/ASR-based data may result in large values of $P(y_i|X_{\setminus i}, x_i)$ due to its inherent long-tail distribution [2], which could result in overconfident predictions. In other words, *Equation 4 provides an upper bound for $P^N(X|Y)$ in the case of random replacement data, facilitating the filtering of noisy samples by setting a confidence threshold.*

**Multi-answer Sample** corresponds to Case 3, where a spelling error can have multiple correct character alternatives. In this case, it is considered a true spelling error ($P_\mathcal{X}(y_i|X_{\setminus i}) = 0$), but there exist multiple corrections other than $x_i$ that are equally valid.

---

[2]The most frequent spelling errors in each character's confusion set in the OCR/ASR-based data constitute 58.7% of the whole misspellings. The percentage is 13.8% for random replacement data

Similar to the analysis of noisy samples, the difference between the two data augmentation methods also relies on $P(y_i|X_{\setminus i}, x_i)$. Further detailed analysis on this matter can be found in Appendix F.

### 3.4 Lessons from The Theoretical Analysis

The theoretical analyses presented above provide a clear explanation for the empirical findings observed in our pilot study. Moreover, they serve as inspiration to utilize the upper-bounded confidence for denoising purposes.

Considering cases 2 and 3, it is important to note that less than 10% of the characters are replaced in the context of data augmentation, $P(y_i|X_{\setminus i}, y_i) \geq 0.9 >> 0.1 \geq P(y_i|X_{\setminus i}, a)$. As long as $P_{\mathcal{X}}(y_i|X_{\setminus i})$ and $P_{\mathcal{X}}(a|X_{\setminus i})$ are of the same order of magnitude, it can be derived that

$$0 < P^N(X|Y) < P^M(X|Y) < P^T(X|Y) = 1 \tag{6}$$

Since the model trained on random replacement data tends to exhibit lower confidence for noisy and multi-answer samples, we can leverage this characteristic to filter out such samples.

The high-level filtering process, guided by the theoretical framework, is illustrated in Figure 2. By using the model's confidence as a threshold, we can effectively identify and remove noisy samples from the dataset, improving the overall quality of the data used for training and evaluation.

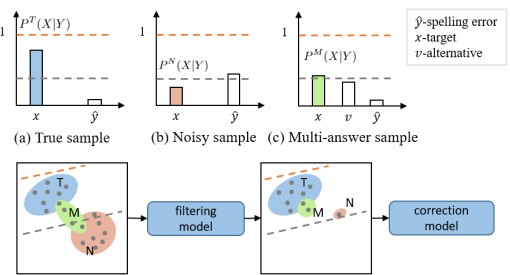

(a) True sample  (b) Noisy sample  (c) Multi-answer sample

(d) Flat representation of sentence space

Figure 2: Conceptual illustration of sample confidence and the filtering process for noisy samples. The upper part demonstrates the variability of model confidence across different samples. The bottom part illustrates the utilization of confidence to identify and filter out noisy samples. The dotted line represents a scalar, while the plane serves as a visual aid for better comprehension.

It is worth noting that multi-answer samples can be real spelling errors (and thus can not be simply treated as noise), but they are rare in the datasets (see Section 6.2). Therefore, removing them from large-scale datasets has a minor impact on the overall performance. Although our primary focus is on eliminating noisy samples, these analyses provide valuable insights into the comprehensive effects of data filtering and its implications for the CSC task itself.

## 4 Approach

### 4.1 The Filtering Strategy

Riding on the analysis above, this paper proposes a filtering model to reduce false spelling errors. We fine-tune BERT on a large-scale news corpus to approach $P(\cdot|X_{\setminus i})$. As for the mapping $\mathcal{F}$, we randomly select 10% of the characters for replacement, and the modified characters are drawn evenly from the confusion set, indicating $P(Y|X) = P(Y'|X)$ for any $y_i, y_i'$ in the confusion set of $x_i$.

The random replacement dataset is used to train our filtering model, which is the Bert-based one introduced in Section 2. Once we obtain a filtering model, we can feed it with data samples of the OCR/ASR-based corpus to be refined. We filter out spelling errors whose recovering confidence of the filtering model is below a certain threshold.

$$y_i' = \begin{cases} y_i & P(X|Y) \geq p \\ x_i & P(X|Y) < p \end{cases} \tag{7}$$

As threshold $p$ increases, more samples will be removed from the training set. In Section 6.5, we will demonstrate the impact of threshold.

## 4.2 THE METHOD PIPELINE

After being processed by the filtering model, the dataset is used to train another Bert-based model with the same architecture as the filtering model, obtaining our final correction model. Algorithm 1 demonstrates the entire process of our approach.

---

**Algorithm 1**

---

1: Train a filtering model $F$ on a large-scale random replacement dataset $D_r$
2: Apply the filtering model $F$ to the OCR/ASR-based dataset $D_o$ and calculate the confidence of spelling errors.
3: Refine $D_o$ according to Equation 7 and get the denoised dataset $D'$
4: Fine-tune a model $M$ for the CSC task with the processed data $D'$

---

## 5 EXPERIMENT SETUP

### 5.1 DATASET

**Auxiliary Training Set.** 9 million sentence pairs are generated with the Chinese News Corpus Xu (2019) by random replacing strategy. The Auxiliary training set is employed to train the filtering model and explore the impact of data volume on the model.

**Training Set.** We use the same training data as previous CSC works Li et al. (2022c); Zhang et al. (2020); Liu et al. (2021); Xu et al. (2021), including the training set from SIGHAN13/14/15 Wu et al. (2013); Yu et al. (2014); Tseng et al. (2015) and the automatic generated data (271k pairs) based on OCR and ASR methods Wang et al. (2018).

**Validation Set.** 1500 pairs from the training set are randomly picked for supervising the training process.

**Test Set.** The test sets from SIGHAN 13/14/15 are employed, and we use the same procedure as previous worksWang et al. (2019); Zhang et al. (2020); Cheng et al. (2020) to transform the text from traditional Chinese to simplified Chinese.

### 5.2 BASELINES

The following baselines are selected: (1) BERT Fine-tuning, BERT model trained on the standard OCR/ASR-based training set; (2) SpellGCN Cheng et al. (2020) employs BERT to extract character representations and constructs two similarity graphs for phonetics and character shapes; (3) PHMO-Spell Huang et al. (2021) extracts phonetic features, character shape features, and context-related semantic features for each character. These features are integrated using an adaptive gate learned through training; (4) DCN Wang et al. (2021) employs an attention-like method to incorporate additional dependency scores for adjacent characters; (5) ECOPO Li et al. (2022c) incorporates an additional contrastive loss to avoid predicting common characters; (6) SCOPE Li et al. (2022a) introduces an auxiliary task of Chinese pronunciation prediction (CPP) to improve CSC; (7) LEAD Li et al. (2022b) also utilizes contrastive learning methods, with negative samples derived from dictionary knowledge and designed based on phonetics, vision, and meaning; (8) Zero-shot ChatGPT (GPT-3.5); (9) Zero-shot ChatGLMDu et al. (2022), a strong Chinese LLM; (10) Finetuned-ChatGLM.

## 6 EXPERIMENT RESULTS

### 6.1 MAIN RESULTS

The results of our method and baselines are shown in Table 2. Our results are obtained by taking the average of five different random seeds. Our approach achieves the highest F1 scores on SIGHAN 13 and SIGHAN 14, significantly surpassing the suboptimal model by margins of 4.1 and 2.4, respectively. We also rank second on SIGHAN 15, 0.3 lower than the best model. We believe achieving the performance by an extremely simple BERT-based CSC model is impressive, highlighting the effectiveness of the data filtering mechanism.

| | SIGHAN13 | | | SIGHAN14 | | | SIGHAN15 | | |
|---|---|---|---|---|---|---|---|---|---|
| | P | R | F1 | P | R | F1 | P | R | F1 |
| SpellGCN | 78.3 | 72.7 | 75.4 | 63.1 | 67.2 | 65.3 | 72.1 | 77.7 | 75.9 |
| PHMOSpell | 99.5 | 74.7 | 85.4 | **81.8** | 63.6 | 71.6 | 88.2 | 68.4 | 77.1 |
| DCN | 84.7 | 77.7 | 81.0 | 65.8 | 68.7 | 67.2 | 74.5 | 78.2 | 76.3 |
| ECOPO | 88.5 | 82.0 | 85.1 | 67.5 | 71.0 | 69.2 | 76.1 | 81.2 | 78.5 |
| SCOPE | 86.3 | **82.4** | 84.3 | 68.6 | **71.5** | 70.1 | 79.2 | **82.3** | **80.7** |
| LEAD | 87.2 | **82.4** | 84.7 | 69.3 | 69.6 | 69.5 | 77.6 | 81.2 | 79.3 |
| ChatGPT | 60.7 | 70.8 | 65.4 | 48.0 | 75.1 | 58.4 | 70.0 | 87.5 | 77.8 |
| ChatGLM | 13.3 | 16.7 | 14.8 | 7.14 | 33.3 | 11.8 | 16.3 | 68.2 | 26.3 |
| ChatGLM-Finetune | 60.0 | 64.2 | 62.0 | 45.2 | 63.8 | 52.9 | 60.0 | 70.6 | 64.9 |
| Ours | **99.7** | 81.2 | **89.5** | 81.7 | 67.7 | **74.0** | **90.1** | 72.5 | 80.4 |

Table 2: The sentence level correction performance on SIGHAN 13/14/15. We use the optimal threshold that achieves the best performance on each dataset. The detailed analysis of confidence thresholding will be presented in Section 6.5. In SIGHAN13, the annotations on "的", "地", "得" are relatively poor, so following the practice of Li et al. (2022c); Xu et al. (2021) we ignore all "的", "地", "得" cases in the evaluation.

Since the CSC task does not involve adding and deleting characters, most previous methods adopt non-autoregressive methods. However, we are interested in how large language models (LLMs) perform in the CSC task due to their powerful learning and generalization abilities. So we further conduct experiments on a proprietary LLM (GPT-3.5) and an open-source LLM (ChatGLM). The reason for unsatisfactory CSC performance for LLMs can be two-fold. On the one hand, they will likely give outputs of different lengths. On the other hand, they may replace some correct words according to their understanding, leading to higher recall and lower precision.

Our data filtering strategy is incorporated into a BERT-based model, so we check its effects by comparing the base model. Table 3 illustrates that our filtering method achieves an all-around improvement on BERT, including higher F1, lower FPR, and lower ECE. We can conclude that training on the refined corpus delivers a performant and well-calibrated CSC model, successfully mitigating over-correction. Therefore, we empirically verify the overall effectiveness of our data filtering strategy.

| Dataset | Model | F1 | FPR | ECE |
|---|---|---|---|---|
| SIGHAN13 | BERT | 80.0 | 37.9 | 0.163 |
| | +Filtering | **89.5** | **6.9** | **0.149** |
| SIGHAN14 | BERT | 72.9 | 17.0 | 0.169 |
| | +Filtering | **74.0** | **14.6** | **0.134** |
| SIGHAN15 | BERT | 78.6 | 15.1 | 0.130 |
| | +Filtering | **80.4** | **7.7** | **0.091** |

Table 3: Performance improvement of our proposed filtering method upon BERT.

Figure 3: Case study of noisy and multi-answer samples.

## 6.2 IDENTIFYING SPECIFIC DATA SAMPLES

Based on the theoretical analysis in Section 3.3, we know that random replacement data can stabilize the model confidence of noisy and multi-answer samples. Here we are keen to see the impacts of our filtering strategy on these samples, but finding that it is non-trivial to accurately identify these samples. Therefore, in this section, we use a heuristic method to roughly find these samples to 1) verify theoretical sample categorization, 2) provide a concrete case study, and 3) support the following experiments about the impacts on noisy and multi-answer samples.

**Noisy Sample Identification.** We replace the modified characters with [MASK] and apply BERT to get the output logits of the mask token. If the ratio of logits corresponding to the characters before and after replacement does not exceed a certain percentage $\lambda_N$, we presume that they are both reasonable in the context, thus we get the dataset $D_N$

**Multi-answer Sample Identification.** Still, we replace the modified characters with [MASK], and we extract the BERT hidden states of the mask token as the representation of the context. If two different characters produce the same misspelling and the cosine similarity of their context representation is over a certain threshold $\lambda_M$, we consider these samples to be multi-answer samples $D_M$. When a context has more than two suitable characters, there is an intersection between $D_N$ and $D_M$. Therefore, we need to remove samples in the intersection to produce the final $D_M$.

We randomly select 3000 samples from the training sets. And we set $\lambda_N = 0.9$ and $\lambda_M = 0.8$ to approximate the sample identification process. 160 noisy samples and 34 multi-answer samples are selected out of 3000. Figure 3 presents two concrete cases, illustrating that the heuristic method can indeed extract noisy and multi-answer samples from the training set. These samples verify our theoretical data categorization and will be further applied to measure the effect of the filtering model in the following experiments.

## 6.3 Other Methods of Corpus Utilization

In this section, we briefly analyze alternative approaches for data utilization. The first approach involves directly combining the two types of datasets (Mixing). The second approach employs the heuristic methods described in noisy sample identification (+H-Filtering). The third approach utilizes the OCR/ASR-based corpus to train a filtering CSC model (S-Filtering). The fourth approach utilizes adaptive training to reduce the weight of negative samples Huang et al. (2020). Note that the heuristic filtering in this experiment primarily focuses on noisy samples for computational efficiency reasons.

| Dataset | Model | P | R | F1 | FPR |
|---------|-------|-----|-----|------|------|
| SIGHAN13 | BERT | 98.3 | 67.4 | 80.0 | 37.9 |
| | Mixing | 99.0 | 74.3 | 84.9 | 22.3 |
| | +H-Filtering | 99.2 | 79.5 | **88.3** | **20.7** |
| | +S-Filtering | 98.4 | 63.9 | 77.5 | 34.5 |
| | +Self-adaptive | 98.7 | 67.8 | 80.4 | 32.1 |
| SIGHAN14 | BERT | 79.2 | 67.5 | 72.9 | 17.0 |
| | Mixing | 80.5 | 67.6 | **73.5** | 15.5 |
| | +H-Filtering | 84.1 | 60.9 | 70.7 | **11.1** |
| | +S-Filtering | 75.7 | 62.9 | 68.7 | 19.4 |
| | +Self-adaptive | 79.6 | 67.4 | 73.0 | 16.3 |
| SIGHAN15 | BERT | 82.8 | 74.7 | 78.6 | 15.1 |
| | Mixing | 86.4 | 73.6 | 79.5 | 11.1 |
| | +H-Filtering | 87.9 | 73.8 | **80.2** | **9.9** |
| | +S-Filtering | 82.5 | 72.3 | 77.1 | 14.9 |
| | +Self-adaptive | 84.6 | 73.8 | 78.8 | 12.1 |

Table 4: Performance of BERT and heuristic/self-filtering method ($\lambda_N = 0.9$) on different datasets.

The results in Table 4 show that the heuristic filtering approach (**+H-Filtering**) improves F1 and leads to better FPR. This verifies our research motivation to denoise corpora. Meantime, **+H-Filtering** lags behind our learnable filtering model in all metrics (refer to Table 3), demonstrating that we purify data more systematically and effectively

The second self-filtering approach is slightly inferior to the baseline model, verifying previous empirical findings and theoretical analysis on the over-confidence of OCR/ASR-based CSC models.

## 6.4 Filtering Effects on Different Data Samples

The heuristic method produces a dataset including both noisy and multi-answer samples, which allows us to measure the effects on these two categories of samples. To corroborate the theoretical analysis, we examine the filtering ratio of these samples by comparing our filter method and self-filtering.

As shown in Figure 4, in line with our expectation, our approach is able to effectively eliminate noisy samples and multi-answer samples. Compared with our method, self-filtering is underperforming in terms of the filtering effect, which explains why the model based on self-filtering gains minor or even negative effects on all the metrics in Table 4.

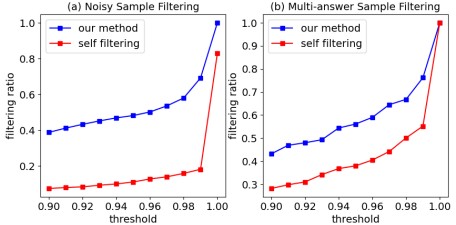
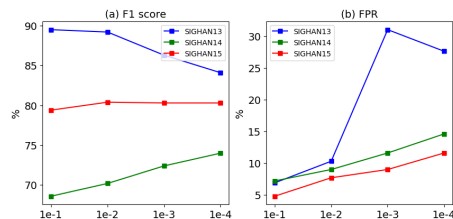

Figure 4: The filtering ratio of noisy samples and multi-answer samples with our method and self-filtering method.

Figure 5: F1 and FPR of the method on three datasets with different filtering thresholds $p$.

## 6.5 EFFECTS OF CONFIDENCE THRESHOLD

Notably, spelling errors in SIGHAN 13/14/15 come in different styles: texts in SIGHAN 13 are mostly in a formal writing style, but texts in SIGHAN 14/15 are in an informal writing style. The effects of our filtering method on these datasets can be different. To observe the influences of the filtering threshold, we experiment with hyper-parameters $p$ of {1e-1,1e-2,1e-3,1e-4} respectively.

According to Figure 5, F1 reduces with decreasing threshold on SIGHAN13 and vice versa on the other two datasets. The reason might be the differences between formal and informal writing styles. Ignoring the outlier, FPR rises as the threshold decreases, which is easy to understand because without filtering the model has a high FPR. The result of ECE is demonstrated in Appendix B. It is optimal at $p = 1e - 2$ on all three datasets. Specifically, if we uniformly use $1e - 2$ as the threshold, our model still outperforms the baselines.

For more auxiliary experiments, refer to Appendix C.

## 7 RELATED WORK

**Chinese spelling correction** (CSC) has made remarkable progress with the help of pre-trained language models (PLMs) such as BERT (Devlin et al., 2018). Fine-tuning over PLMs became mainstream solutions Zhang et al. (2020); Nguyen et al. (2021); Bao et al. (2020). Furthermore, more improvements to CSC are achieved by incorporating phonological and visual information into PLMs Jin et al. (2014); Cheng et al. (2020); Xu et al. (2021); Zhang et al. (2021b); Huang et al. (2021).

**Data denoising** is a general concern as noisy labels severely degrade the generalization of a deep learning model Zhang et al. (2021a). In addition to regularization and loss design, some works directly conduct sample selection. Assigning weights to potentially incorrect samples is a kind of approach Jiang et al. (2018); Ren et al. (2018). Usually, the weights are extremely low compared to those of normal samples. Another way is to filter out potentially wrong samples directly Tam Nguyen et al. (2019), which means their weights are either zero or one. In this paper, we also drop the false spelling errors, considering that we have an almost infinite training set.

## 8 CONCLUSION

We propose a simple, efficient, and interpretable data filtering method to purify Chinese Spelling Correction (CSC) corpora. We empirically reveal and theoretically prove the promising calibration characteristic of CSC models trained on random replacement datasets. Using a well-calibrated CSC model to filter the OCR/ASR-based corpora, we learn a final CSC model that integrates the strong generalization performance from OSC/ASR-based data and the trustful model confidence from random replacement data. Our method impressively achieves state-of-the-art performance on SIGHAN 13/14/15 and significantly alleviates over-corrections.

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

| Augmentation | SIGHAN13 | | | | SIGHAN14 | | | | SIGHAN15 | | | |
|---|---|---|---|---|---|---|---|---|---|---|---|---|
| | P | R | F1↑ | FPR↓ | P | R | F1↑ | FPR↓ | P | R | F1↑ | FPR↓ |
| Random | **99.1** | 54.7 | 70.5 | **17.2** | 77.2 | 39.0 | 51.9 | **11.1** | **87.3** | 50.7 | 64.2 | **7.2** |
| OCR/ASR | 98.4 | **71.3** | **82.7** | 37.9 | **79.7** | **72.7** | **76.1** | 17.7 | 83.5 | **77.3** | **80.3** | 14.9 |

Table 5: Performance of BERT models trained on differently augmented data. The metrics are Precision(P), Recall(R), F1-score(F), and sentence-level False Positive Rate(FPR). The model trained with OCR/ASR-based data has a higher F1-score at the cost of more erroneous judgement.

## A  PRELIMINARIES: CALIBRATED CONFIDENCE ESTIMATION

Calibration plays a crucial role in enhancing the interpretability of models, primarily because humans have a tendency to associate confidence with probability. To establish a formal understanding, it is essential to define the concept of perfect calibration. We expect the perfect calibration to adhere to the following criterion:

$$P(\hat{Y} = Y|\hat{P} = p) = p, \forall p \in [0, 1] \tag{8}$$

Here, $\hat{Y}$ and $\hat{P}$ represent the predicted labels and corresponding probabilities, while $Y$ denotes the ground truth. This formulation ensures that the predicted probabilities closely match the actual probabilities assigned to the outcomes.

To quantitatively evaluate the calibration performance, we can employ a scalar summary statistic known as the Expected Calibration Error (ECE) Guo et al. (2017). The ECE can be defined as follows:

$$ECE = \mathbb{E}_{\hat{P}}[|P(\hat{Y} = Y|\hat{P} = p) - p|] \tag{9}$$

In practical calculations, the accuracy of samples falling within a specific prediction probability interval is often used to approximate the value of $p$. This approach allows for a practical assessment of calibration performance.

## B  SUPPLEMENTARY EXPERIMENTAL RESULTS FOR DIFFERENT METRICS ON MULTIPLE DATASETS

The figure presented in Section 1 is derived from the SIGHAN13 dataset, providing a visual representation of the observed results. However, it is important to note that conducting experiments on other widely recognized datasets can further validate and strengthen the findings. In Table 5, we showcase the outcomes of experiments performed on these additional datasets, demonstrating the differences between the two types of augmentation methods.

The results of the Expected Calibration Error (ECE) with varying filtering thresholds are visually represented in Figure 6, which serves as a valuable supplement to the discussions in Section 6.5.

## C  EFFECTS OF DATA VOLUME

| Corpus Size | SIGHAN13 | | SIGHAN14 | | SIGHAN15 | |
|---|---|---|---|---|---|---|
| | F1 | FPR | F1 | FPR | F1 | FPR |
| 5k | 84.5 | **6.9** | 59.6 | 8.9 | 70.7 | 6.1 |
| 10k | 82.9 | **6.9** | 59.6 | **8.5** | 70.2 | **5.9** |
| 100k | 83.7 | 10.3 | 60.0 | 9.2 | 69.6 | 7.5 |
| 200k | 83.5 | 10.3 | 60.6 | 10.1 | 71.2 | 7.9 |
| 400k | 84.1 | 10.3 | 62.8 | 9.6 | 71.5 | 7.5 |
| 2m | 86.7 | 13.8 | 65.7 | 10.0 | 75.1 | 8.1 |
| 9m | **89.2** | 10.3 | **70.2** | 9.0 | **80.4** | 7.7 |

Table 6: Experimental results on the effects of pre-training corpus size.

Our auxiliary experiments have been centered around $P_{\mathcal{F}}(\hat{y}_i|x_{\setminus i}, x_i)$ in Equation 4. We take $P(v|x_{\setminus i})$ as a default constant. However, a small

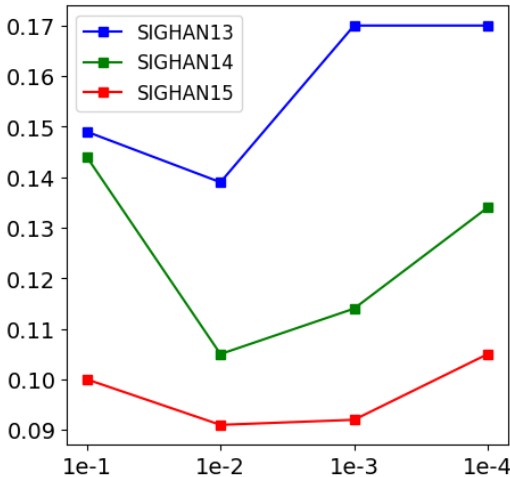

Figure 6: ECE of the method on three datasets with different filtering thresholds $p$.

corpus size is likely to lead to estimation bias on $P(v|x_{\backslash i})$ when calculating the confidence, we explore how large a pre-training sample size would be more appropriate.

We set the filtering thresholds $p = 1e - 2$ and experiment on diverse sizes of the dataset for the pre-trained filtering model. Table 6 shows that the F1-score of the model gradually increases as the corpus grows, and the FPR remains in a stable interval. In order to achieve better model performance and maintain the stability of $P(v|x_{\backslash i})$, a million-data volume is necessary.

## D  BAYESIAN INFERENCE OF MODEL CONFIDENCE

This section presents the derivation of Equation 2, which builds upon the assumptions outlined in Section 3. By applying the Bayesian formula, we can express the equation as follows:

$$
\begin{aligned}
P(X|Y) &= P(Y|X) \cdot \frac{P_{\mathcal{X}}(X)}{P_{\mathcal{Y}}(Y)} \\
&= P(y_i|X) \cdot \frac{P_{\mathcal{X}}(x_i|X_{\backslash i})P_{\mathcal{X}}(X_{\backslash i})}{P_{\mathcal{Y}}(y_i|Y_{\backslash i})P_{\mathcal{Y}}(Y_{\backslash i})} \\
&= P(y_i|X) \cdot \frac{P_{\mathcal{X}}(x_i|X_{\backslash i})}{P_{\mathcal{Y}}(y_i|X_{\backslash i})}
\end{aligned}
\tag{10}
$$

In the formulation, $P(\cdot|X_{\backslash i})$ represents the conditional probability of a character given the context $X_{\backslash i}$. Since $P_{\mathcal{Y}}$ is influenced by the augmentation method $\mathcal{F}$, we expand $P_{\mathcal{Y}}(y_i|X_{\backslash i})$ as follows:

$$
P_{\mathcal{Y}}(y_i|X_{\backslash i}) = \sum_{v \in \mathcal{V}} P(y_i|X_{\backslash i}, v)P_{\mathcal{X}}(v|X_{\backslash i})
\tag{11}
$$

And the Eq. 10 can be expressed as Eq. 2

## E  NOISY SAMPLE CONFIDENCE SUPPLEMENT

In Section 3, we focus on providing confidence estimates specifically in the case of two correct characters for the same context. The complete formula for this scenario is as follows:

$$P^N(X|Y) = \frac{1}{1 + \frac{P_{\mathcal{X}}(y_i|X_{\backslash i})}{P_{\mathcal{X}}(x_i|X_{\backslash i})} \frac{P(y_i|X_{\backslash i}, y_i)}{P(y_i|X_{\backslash i}, x_i)} + \sigma(X, Y)}$$

$$\sigma(X, Y) = \sum_{v \in \mathcal{V} \backslash \{x_i, y_i\}} \frac{P_{\mathcal{X}}(v|X_{\backslash i})}{P_{\mathcal{X}}(x_i|X_{\backslash i})} \cdot \frac{P(v|X_{\backslash i}, v)}{P(v|X_{\backslash i}, x_i)} \quad (12)$$

Here, $\sigma(X, Y)$ represents a non-negative value that depends on the vocabulary $\mathcal{V}$. It is worth noting that if $x_i$ and $y_i$ are the only two suitable characters given the context $X_{\backslash i}$, then $\sigma(X, Y) = 0$. Consequently, Equation 4 already provides an upper bound in this case.

## F  QUANTITATIVE ANALYSIS OF MODEL CONFIDENCE

Previous studies have commonly utilized a random selection of 10% of the characters to simulate the distribution of human misspellings $\mathcal{Y}$. In line with this established approach, we follow the same methodology in this paper. Accordingly, we assign the following probabilities: $P(x_i|X_{\backslash i}, x_i) = 0.9$ and $P(y_i|X_{\backslash i}, x_i) \leq 0.1$, where $y_i \neq x_i$.

Additionally, we make the assumption that for any two characters $u$ and $v$ suitable for a given context, the ratio $\frac{P_{\mathcal{X}}(u|X_{\backslash i})}{P_{\mathcal{X}}(v|X_{\backslash i})} \geq a$. With these assumptions in place, we can establish a numerical upper bound for Equation 12:

$$
\begin{aligned}
P^N(X|Y) &= \frac{1}{1 + \frac{P_{\mathcal{X}}(y_i|X_{\backslash i})}{P_{\mathcal{X}}(x_i|X_{\backslash i})} \frac{P(y_i|X_{\backslash i}, y_i)}{P(y_i|X_{\backslash i}, x_i)} + \sigma(X, Y)} \\
&\leq \frac{1}{1 + \frac{P_{\mathcal{X}}(y_i|X_{\backslash i})}{P_{\mathcal{X}}(x_i|X_{\backslash i})} \cdot \frac{P(y_i|X_{\backslash i}, y_i)}{P(y_i|X_{\backslash i}, x_i)}} \\
&\leq \frac{1}{1 + 9a}
\end{aligned}
\quad (13)
$$

This implies a low model confidence when taking a reasonable $a = 0.1$ and $P^N(X|Y) \leq 0.53$, indicating that noisy samples can be easily filtered out by a pre-trained model regardless of the choice of $\mathcal{F}$. As mentioned in Section 3, due to the existence of a long-tailed distribution for the OCR method, there exists a $y_i$ that gives $P^N$ a larger upper bound compared to random replacement.

Handling multi-answer samples presents a more complex challenge. When $\mathcal{F}$ represents a mapping of uniformly sampling misspellings from a confusion set, we can derive that $\forall u, v \in \mathcal{V}x, \frac{P(y_i|x\backslash i, u)}{P(y_i|X_{\backslash i}, v)} = \frac{|C_u|}{|C_v|}$, where $C_v$ denotes the confusion set of character $v$. In this case, we assume that $\frac{|C_u|}{|C_v|} \geq b$. Consequently, we can establish a numerical upper bound for Equation 5:

$$
\begin{aligned}
P^M(X|Y) &= \frac{1}{1 + \sum_{v \in \mathcal{V}} \frac{P_{\mathcal{X}}(v|X_{\backslash i})}{P_{\mathcal{X}}(x_i|X_{\backslash i})} \frac{P(y_i|X_{\backslash i}, v)}{P(y_i|X_{\backslash i}, x_i)}} \\
&\leq \frac{1}{1 + \sum_{v \in \mathcal{V}} ab} \\
&\leq \frac{1}{1 + ab}
\end{aligned}
\quad (14)
$$

Here, $a$ and $b$ represent lower bounds for the ratio, and in practice, they are typically small values. Let's assume $a = 0.1$ and $b = 0.5$, we find that $P^N(X|Y) \leq 0.96$. Consequently, selecting multi-answer samples is considerably more challenging than dealing with noisy samples, especially when the pre-trained model fails to achieve the theoretical upper bound of confidence. Furthermore, the long-tailed distribution observed in the OCR method results in a larger potential value for $b$, thereby further intensifying the challenge of differentiation.

## G    LIMITATIONS

The main limitation of our approach is that we need to search for the best threshold for different datasets, even though a rough threshold (e.g., $1e - 2$) can also bring significant performance improvement across all datasets. On the one hand, this phenomenon is natural since different datasets commonly have their unique distribution. On the other hand, it will not affect the application of our method in practice too much, since the effort of threshold searching is tolerable, and we typically face similar data distribution (e.g., in a specific domain) in real-world scenarios.

## H    IMPLEMENTATION DETAILS

Most hyperparameters are shared across all experiments to avoid dataset-specific tuning. Based on the repository of Transformers, We train our model using AdamW optimizer for 10 epochs with a learning rate decay of 5e-5, and batch size is set to 50 for each experiment. All experiments were performed using 4 Nvidia A100 GPUs.

