# OpenReview forum: "Refining Corpora from a Model Calibration Perspective for Chinese Spelling Correction"
_ICLR.cc/2024/Conference — Submitted to ICLR 2024_

### Official Review · Reviewer_DKJd · 2023-10-17

**Soundness:** 3 good
**Presentation:** 3 good
**Contribution:** 2 fair
**Rating:** 5
**Confidence:** 2

**Summary:**

The paper proposes an amazingly simple formula (eq 7) for combining observations with pseudo-labels.  The proposed method is shown to be effective for Chinese spelling correction.

**Strengths:**

The method is amazingly simple, and appears to be effective.

**Weaknesses:**

The approach seems too good to be true.

There is a huge literature on pseudo-labels, self-training, co-training, EM, etc.  These methods have many applications that go way beyond Chinese spelling correction.

Here are a few highly cited examples:

https://arxiv.org/pdf/1905.02249.pdf
https://arxiv.org/pdf/1908.02983.pdf
https://www.cs.cmu.edu/~avrim/Papers/cotrain.pdf

There are a number of baselines in table 4, but I found it difficult to understand what each of them do.  I wonder if the description of the method could be shortened in order to make more space available for related work and baselines.

Many readers may not appreciate the challenges in spelling correction for Chinese.  I might start with a discussion like Jurafsky's book (https://web.stanford.edu/~jurafsky/slp3/B.pdf), where they have a language model and a channel model.  You assume errors are just one character for one.  Chinese may be simpler than English in that respect.

As for the channel model, I'm surprised that you have just two models in mind: (1) random and (2) similar in OCR space.  I might have thought of some others like (3) similar in pinyin space, (4) dependencies involving dialects, (5) dependencies involving input methods (6) similar in distribution.

It is well known that spelling correction depends a lot on the context.  We should expect to see very different errors depending on the keyboard.  Typos are different when the user is on a laptop or a phone.  Within phones, there are different keyboards.

The method of estimating the channel model is somewhat similar to the proposed method here.  They used a boot strapping method where they started with a very simple method to find typos that had just one reasonable correction.  They found  enough of those that they could then estimate confusion matrices.  That probably wouldn't work for Chinese, but it isn't that different from your proposal of training on cases where the probability of the correction is reasonably high.

**Questions:**

Can you generalize your work so the paper could be of interest to a larger community of people interested in pseudo-labels, self-training, co-training, etc.?

Can you say more about the baselines?

Can you say more about how spelling correction is different in Chinese from spelling correction is other languages?

---

> ### Author Response · Authors · 2023-11-23
>
> Thanks for your constructive and insightful comment.
>
> **Question 1: Generalization in the fields of pseudo-labels, self-training, co-training, etc.**
>
> Our findings indicate that models trained on OCR/ASR data perform well but tend to be overconfident. Analysis reveals that OCR/ASR methods often result in false spelling errors. Conversely, models trained on randomly replaced data exhibit better calibration. This enables the correction of biases in the OCR/ASR dataset.
>
> Our focus is on using a well-calibrated model to correct biased datasets. Extending the method to other domains is worthy of exploration, considering that self-training and pseudo-labeling can also lead to overconfident classifications. We can add some relavant discussions on these topics.
>
> **Question 2: Introduction to baseline work**
>
> As mentioned in the review, there are several optional channel models, with many current works based on phonetics, character shapes, and other aspects.
>
> - SpellGCN: It employs BERT to extract character representations and constructs two similarity graphs for phonetics and character shapes. These graphs capture interdependencies between characters and generate a target vector with information about interactions among similar characters.
>
> - PHMOSpell: It extracts phonetic features, character shape features, and context-related semantic features for each character. These features are integrated using an adaptive gate learned through training. Phonetic extraction includes mel-spectrograms, while character shapes are encoded using image encoding.
>
> - DCN: Recognizing that connected characters can form words, it employs an attention-like method to incorporate additional dependency scores for adjacent characters.
>
> - ECOPO: It treats correct characters as positive samples and characters that the model tends to predict incorrectly (often common characters) as negative samples. It incorporates an additional contrastive loss.
>
> - SCOPE: It introduces pinyin (phonetic transcription) as an additional supervision for prediction.
>
> - LEAD: It also utilizes contrastive learning methods, with negative samples derived from dictionary knowledge and designed based on phonetics, vision, and meaning.
>
>
> Since the paper's focus is not on model architecture design, only basic models were used. Additional descriptions of the baseline models will be provided in the revised paper.
>
> **Question 3: How spelling correction is different in Chinese from spelling correction is other languages**
>
> Firstly, there is a granularity issue. In comparison to English, Chinese characters have a significantly larger scale. English has 26 characters, while Chinese has tens of thousands of characters, each with nuanced semantics. Chinese misspellings can be likened to modifying a single character in an English word, resulting in a distinct word with a different meaning.
>
> Secondly, Chinese input methods can be classified into two categories: shape-based encoding and pinyin encoding. Both types have a substantial user base, making phonetics and character shapes crucial factors in Chinese spelling correction.

---

### Official Review · Reviewer_65Mo · 2023-10-30

**Soundness:** 3 good
**Presentation:** 3 good
**Contribution:** 3 good
**Rating:** 6
**Confidence:** 4

**Summary:**

This paper centers its attention on addressing the calibration problem within the realm of Chinese Spelling Check (CSC). Due to the lack of large corpus in the field of CSC, two data augmentation methods of random replacement and OCR/ASR-based generation is proposed to generate large-scale corpora. These methods introduce noise into the data, which subsequently causes over-correction. The authors analyze the calibration of the CSC models trained in the two corpora, and observe that random replacement results in better-calibrated CSC models. The authors then propose a corpus refining strategy to filter OCR/ASR-based data. Utilizing a BERT-based model trained on this refined corpus, the authors achieve commendable performance on CSC benchmarks, affirming the efficacy of the proposed method in mitigating over-correction.

**Strengths:**

1.	The paper makes a valuable contribution by highlighting the differences in generalization performance between OCR/ASR-based and random replacement data augmentation techniques. This insight has the potential to inspire further research in the CSC domain.
2.	The paper proposes a novel data filtering method after carefully observation of the two data augmentations in CSC. The method effectively filters the noisy examples, and the model trained on the refined corpus can achieve impressive performance.
3.	The motivation behind the research is clearly justified and based on empirical observations. The proposed methodology is presented in a comprehensible manner, making it suitable for adaptation to other models in the field.

**Weaknesses:**

1.	The statistical data presented in Table 2 appears to contain an error, as the reported F1 score for the SIGHAN 15 dataset does not align with the provided precision and recall values. This discrepancy requires clarification.
2.	Missing citations of ChatGLM. It remains unknown that which version of ChatGLM (ChatGLM or ChatGLM2?) is used in this paper. A more comprehensive citation and elaboration on the fine-tuning procedure are needed.
3.	Excessive white space around tables, maybe the layout can be adjusted.

**Questions:**

1.	How do you design the prompt of the LLMs to generate the corrected results? Have you tried other templates to generate?
2.	The issue of varying output lengths generated by LLMs is mentioned. Could you provide additional information on the strategies employed to mitigate this problem and ensure consistent results?

---

> ### Author Response · Authors · 2023-11-23
>
> Thanks for your constructive and insightful comment.
>
> **Question 1: Text and citation issues**
>
> Thank you for your careful reading. We have addressed the omission in Table 2. Additionally, it should be noted that the experiments involving LLM are conducted using ChatGLM-6B rather than ChatGLM2. We have included the appropriate citation.
>
> **Question 2: The prompt of the LLMs to generate the corrected results**
>
> We employed a variety of prompts, such as “Chinese spelling correction task, involving only character replacements, without additions, deletions, or extra output.” Yet none managed to consistently yield successful task execution across all samples.
>
> **Question 3: Additional information on the strategies employed to varying output lengths generated by LLMs**
>
> LLMs often make synonym substitutions, which are irrelevant for Chinese spelling correction. In our experiments, we removed samples that generated lengths that did not meet the requirements. However, even among samples with the same length, there might still be instances of synonym substitutions, leading to LLMs exhibiting higher recall and lower precision. Although reproducing these occurrences is challenging, case studies indicate that they are infrequent.

---

### Official Review · Reviewer_7Y3U · 2023-10-31

**Soundness:** 3 good
**Presentation:** 3 good
**Contribution:** 2 fair
**Rating:** 6
**Confidence:** 2

**Summary:**

The paper gives a clear picture of 2 mainstream techniques to create mis-spelling dataset for chinese characters, Random Replacement and OCR/ASR. However, each approach of corruption brings its own inherent bias that may lead to problems in training. For example, OCR/ASR way of corrupting characters gives very calibrated scores, as shown as Figure-1, while the Models trained on Random Replacement data has less calibration issues. The authors propose a recipe to train on both Random Replacement and OCR/ASR data, but with some scheduling and post-process steps, which include prioritize Random Replacement training to learn calibration, and then do OCR/ASR training to improve over-all performance.

There is a typo in the Table-2 SIGHAN15, but overall, the proposed method achieves SOTA performance on benchmark datasets.

**Strengths:**

The strength of the paper lies in its clear presentation of the problem, robust experiment designs, and strong performance on benchmark datasets. The problem of calibration vs overall performance is laid out as the bottleneck, and the authors offer a recipe to combine the two methods.

First, the paper is well-written.

The illustration of the problem is made clear by Figure-1, where Calibration on both models are weak, but the OCR/ASR is much worse. However, the overall performance shows OCR methods is better across-the-board.

The final results is validated on 3 benchmark Chinese spelling datasets, with 6 existing spelling model benchmarks, and 3 Generative AI benchmarks. The performance gain is significant.

Lastly, the issue of mis-spelling in Chinese is a practical and important problem. It will be a waste of resources to rely on a 10 billion parameter ChatGPT to do it on everyday use cases, though the paper shows not so great Zero-Shot performance by ChatGPT.

**Weaknesses:**

The authors made direct reference of "Random Replacement" and "OCR/ASR" methods as the two mainstream way to construct mis-spelling datasets. The fact that we can train on data created by both methods isn't a source of novelty.

The paper uses a model trained on Random Replacement to filter/"refine" OCR/ASR corpus is kind of interesting, but may be a step that introduces another layer of inductive bias.

The author mentions that "We use the optimal threshold that achieves the best performance on each dataset." This is not a good idea, because it runs the risk of leaking test data to model developers. The final performance gain should be reported on 1 uniform threshold(might be proportion if different datasets have different absolute values) determined by running ablation on a separated dev-set . In that case, we can be sure that whatever gain that we see in Table-2 is from the novel method.

**Questions:**

1. Are there other works that combine Random Replacement and OCR methods in training?

2. Is there a leak of test-set when determining the threshold for filtering?

3. Is there a reason to choose different threshold for each dataset? Can you report Table-2 using one uniform cut-off, and report the dev-set/test-set performance separately?

---

> ### Author Response · Authors · 2023-11-23
>
> Thanks for your constructive and insightful comment.
>
> **Question 1: Other works that combine Random Replacement and OCR methods in training**
>
> To the best of our knowledge, there are no other works that combine Random Replacement and OCR/ASR methods in the field of Chinese spelling correction. In fact, there is no recognized random replacement dataset. The motivation behind our approach extends beyond simple data combination; it stems from our observation that existing OCR/ASR datasets often result in overconfidence of the model due to some false spelling errors. Our findings demonstrate that models trained on random replacement data exhibit better calibration (as illustrated in Figure 1) and suitable to train a filtering model. Thus, we construct a comprehensive random replacement dataset.
>
> **Question 2: Is there a leak of test-set when determining the threshold for filtering**
>
> In our paper, the data filtering we discussed involves utilizing models trained on random replacement data to filter the SIGHAN training set. It is important to note that the SIGHAN dataset consists of distinct training, validation, and testing sets. Aside from employing a reduced training data, our approach aligns with other reference papers that utilize the same dataset. There is no concern of data leakage in our methodology.
>
> **Question 3: Is there a reason to choose different threshold for each dataset**
>
> We acknowledge that the use of varying thresholds may raise concerns. However, in our paper, we address this issue by providing detailed explanations and conducting comparative experiments.
>
> Firstly, our threshold search is designed to be relatively coarse, with values of 1e-1, 1e-2, 1e-3, and 1e-4. This choice is made to avoid the complexity of an exhaustive threshold search and to ensure the usability of our method.
>
> Secondly, in section 6.5, we thoroughly analyzed the impact of different datasets on the results, as demonstrated in Figure 5. Notably, the effect of threshold variations on the SIGHAN 13 dataset differs from the other two datasets. We recognized the concern that using the same threshold in Table 2 might weaken the persuasiveness of our method. However, we are more than willing to provide individual results for the three datasets at a threshold of 1e-2 upon request.
>
> | p=1e-2 | P   | R   | F1  | FPR |
> | --- | --- | --- | --- | --- |
> | SIGHAN13 | 99.5±(0.03) | 80.1±(0.58) | 88.8±(0.37) | 10.7±(0.96) |
> | SIGHAN14 | 85.8±(0.82) | 59.0±(0.45) | 69.9±(0.48) | 9.1±(0.23) |
> | SIGHAN15 | 89.9±(1.10) | 72.1±(0.39) | 80.0±(0.64) | 7.8±(0.16) |
>
> We achieved performance surpassing the baseline across various thresholds, and at the optimal threshold, we outperformed models that employed complex designs.

---

### Meta-Review · Area_Chair_J6vN · 2023-12-11

**Metareview:**

This paper presents to techniques to generate data for training models to correct Chinese spelling errors. Random Replacement and OCR/ASR. Both of these methods are used to generate training dats independently and then a Bard based model is trained to correct the spelling mistakes. Since these automatic methods produce noisy training data, the models do over-correction, and the authors this try to fix the noisy training data. The authors analyze the calibration of the spelling correction models trained in the two corpora, and observe that random replacement results in better-calibrated models. Thus the authors propose a corpus cleaning strategy to filter OCR/ASR-based data. Utilizing a BERT-based model trained on this refined corpus, the authors achieve SOTA results on some datasets. The paper is well written and provides theoretical understanding of why the models are working.

**Justification For Why Not Higher Score:**

The technique in itself is not anything new, and hence its not surprising that it works. I find the contribution of the paper to not be very strong.

**Justification For Why Not Lower Score:**

n/a

---

### Decision · Program_Chairs · 2024-01-16

Reject